# Epidemiology and Transmitted HIV-1 Drug Resistance among Treatment-Naïve Individuals in Israel, 2010–2018

**DOI:** 10.3390/v14010071

**Published:** 2021-12-31

**Authors:** Tali Wagner, Neta S. Zuckerman, Tami Halperin, Daniel Chemtob, Itzchak Levy, Daniel Elbirt, Eduardo Shachar, Karen Olshtain-Pops, Hila Elinav, Michal Chowers, Valery Itsomin, Klaris Riesenberg, Marina Wax, Rachel Shirazi, Yael Gozlan, Natasha Matus, Shirley Girshengorn, Rotem Marom, Ella Mendelson, Dan Turner, Orna Mor

**Affiliations:** 1Sackler Faculty of Medicine, Tel-Aviv University, Tel Aviv 6997801, Israel; tali.vagner@sheba.health.gov.il (T.W.); Itsik.Levy@sheba.health.gov.il (I.L.); chowersm@post.tau.ac.il (M.C.); ella.mendelson@sheba.health.gov.il (E.M.); dant@tlvmc.gov.il (D.T.); 2Chaim Sheba Medical Center, National HIV-1 and Viral Hepatitis Reference Laboratory, Ramat Gan 5262112, Israel; neta.zuckerman@sheba.health.gov.il (N.S.Z.); Marina.Wax@sheba.health.gov.il (M.W.); Rachel.Shirazi@sheba.health.gov.il (R.S.); Yael.Gozlan@sheba.health.gov.il (Y.G.); 3Tel-Aviv Sourasky Medical Center, Crusaid Kobler AIDS Center, Tel Aviv 6423906, Israel; tamihal@tlvmc.gov.il (T.H.); hivlab@tlvmc.gov.il (N.M.); shirleygi@tlvmc.gov.il (S.G.); rotemma@tlvmc.gov.il (R.M.); 4Faculty of Medicine, Braun School of Public Health & Community Medicine, Hebrew University-Hadassah Medical School, Jerusalem 9112102, Israel; daniel.chemtob@moh.gov.il (D.C.); ElbirtDa@clalit.org.il (D.E.); hila.elinav@gmail.com (H.E.); 5Tuberculosis and AIDS Department, Ministry of Health, Jerusalem 9101002, Israel; 6Chaim Sheba Medical Center, Infectious Disease Unit, Ramat Gan 5262112, Israel; 7Immunology, Kaplan Medical Center, Rehovot 76100, Israel; 8Immunology Unit, Rambam Health Care Campus, Haifa 3109601, Israel; ed_shahar@rambam.health.gov.il; 9Rappaport Faculty of Medicine, Institute of Technology, Technion, Haifa 3200003, Israel; 10Hadassah Medical Center, Jerusalem 9112102, Israel; Kerenop@hadassah.org.il; 11Infectious Diseases, Meir Medical Center, Kfar Saba 4428164, Israel; 12Hillel Yaffe Medical Center, Hadera 38100, Israel; ValeryI@hy.health.gov.il; 13Faculty of Health Sciences, Goldman Medical School, Ben-Gurion University of the Negev, Beer-Sheva 8410501, Israel; klaris@bgu.ac.il; 14Soroka Medical Center, Infectious Disease Institute, Beer-Sheva 84101, Israel

**Keywords:** epidemiology, people living with HIV-1(PLHIV), HIV-1 spread, transmitted drug-resistance mutations (TDRM)

## Abstract

Despite the low prevalence of HIV-1 in Israel, continuous waves of immigration may have impacted the local epidemic. We characterized all people diagnosed with HIV-1 in Israel in 2010–2018. The demographics and clinical data of all individuals (*n* = 3639) newly diagnosed with HIV-1 were retrieved. Subtypes, transmitted drug-resistance mutations (TDRM), and phylogenetic relations, were determined in >50% of them. In 39.1%, HIV-1 transmission was through heterosexual contact; 34.3% were men who have sex with men (MSM); and 10.4% were people who inject drugs. Many (>65%) were immigrants. Israeli-born individuals were mostly (78.3%) MSM, whereas only 9% of those born in Sub-Saharan Africa (SSA), Eastern Europe and Central Asia (EEU/CA), were MSM. The proportion of individuals from SSA decreased through the years 2010–2018 (21.1% in 2010–2012; 16.8% in 2016–2018) whereas those from EEU/CA increased significantly (21% in 2010–2012; 27.8% in 2016–2018, *p* < 0.001). TDRM were identified in 12.1%; 3.7, 3.3 and 6.6% had protease inhibitors (PI), nucleotide reverse transcriptase inhibitors (NRTI), and non-nucleoside reverse transcriptase inhibitors (NNRTI) TDRM, respectively, with the overall proportion remaining stable in the studied years. None had integrase TDRM. Subtype B was present in 43.9%, subtype A in 25.2% (A6 in 22.8 and A1 in 2.4%) and subtype C in 17.1% of individuals. Most MSM had subtype B. Subtype C carriers formed small clusters (with one unexpected MSM cluster), A1 formed a cluster mainly of locally-born patients with NNRTI mutations, and A6 formed a looser cluster of individuals mainly from EEU. Israelis, <50 years old, carrying A1, had the highest risk for having TDRM. In conclusion, an increase in immigrants from EEU/CA and a decrease in those from SSA characterized the HIV-1 epidemic in 2010–2018. Baseline resistance testing should still be recommended to identify TDRM, and improve surveillance and care.

## 1. Introduction

Israel is a country of immigration, with a relatively low prevalence of HIV-1 compared with other Western countries (0.1%) [1]. Before 2010, the Israeli Ministry of Health (MoH) reported 6579 HIV-1 cases, of whom 41.3% originated from countries with generalized epidemics in Sub-Saharan Africa, SSA. The transmission groups of the 6579 HIV-1 cases were men who have sex with men (MSM) (21.3%), and people who use injection drugs (PWID) (13.4%). While most individuals from SSA were considered to be infected though heterosexual transmission, only 12.1% of those originating from countries other than SSA were considered to be infected through heterosexual transmission [1]. As immigration continues, and several waves of immigration from countries in SSA and Eastern Europe and Central Asia (EEU/CA) have been documented in the last decade, the landscape of HIV-1 positive individuals in Israel after 2010 may have changed [2,3,4,5].

This change may have affected the circulating viral subtypes. HIV-1 in Israel started with a subtype-B epidemic among MSM, and following the mass migration from Ethiopia and EEU in the 1980s and 1990s, subtypes C and A were identified [6,7]. Recently, a new classification of subtype A was proposed, with subtype A subdivided into six sub-subtypes (A1–A4, A6, A7) [8]. Furthermore a new sub-subtype was also detected: sub-subtype A8 [9]. A1 sub-subtype is predominantly found in SSA, and A6 is unique to regions in the EEU [10]. Previously, any of these A sub-subtypes were not reported in Israel [7].

HIV-1 drug-resistance testing for people with HIV-1 (PLHIV) is suggested to guide the selection of the initial antiretroviral therapy regimen. In Israel, pretreatment protease (PR) and reverse transcriptase (RT) drug-resistance mutations were previously observed in 10–14% of cases [11,12], and the Ministry of Health authorized PR and RT resistance testing to all newly diagnosed individuals. As integrase pretreatment mutations have rarely been reported [13], baseline integrase resistance in Israel was not previously suggested or reported.

The last nationwide study [1] that analyzed demographic and clinical determinants of HIV-1 infections studied all newly diagnosed individuals identified before 2010. Here, we focused on the period 2010–2018. We describe the major epidemiological characteristics of all newly diagnosed PLHIV, and also the spread of transmitted drug-resistance mutations (TDRM, based on the surveillance list that contains 93 RT and PR mutations, by Bennet, 2009, [14]). We also present the prevalence of any HIV-1 drug-resistance mutation (HIVdrm, according to the Stanford HIV drug-resistance database, [15]) to enable comparison to other reports that use the latter approach. Analysis of baseline resistance is of relevance especially in the current era of treatment simplification, and of shifting to dual therapy regimens [16].

## 2. Materials and Methods

### 2.1. Patients

A retrospective multi-site cross-sectional cohort study of data retrieved from the National HIV Reference Laboratory (NHRL) of the MoH was cross-matched with data from the National HIV Registry managed by the Department of Tuberculosis and AIDS of the MoH, and with clinical data from the HIV regional centers. All PLHIV above 18 years old, diagnosed between January 2010 and December 2018, were included. Children < 18 years, adults diagnosed in years other than 2010–2018, or PLHIV who were treated, were excluded. Demographic (sex, age, country of birth, possible mode of HIV-1 transmission), clinical (year of HIV-1 diagnosis, HIV-1 viral load, CD4 cell counts) and molecular data (HIV-1 subtype, TDRM and HIVdrm from available nucleotide sequences) were collected. Country of birth was categorized to SSA, EEU/CA, Western and Central Europe and North America, WCEU/NA, Israel and Other (details in Appendix A).

### 2.2. Genotyping

In Israel, HIV-1 resistance analysis is performed in two laboratories: the NHRL and the HIV laboratory of the Sourasky Medical Center. All available partial protease (PR, codons 4-99), reverse transcriptase (RT, codons 38-247) and integrase (IN, codons 37-280) sequences from the first blood sample taken less than six months following initial HIV-1 diagnosis (single sequence per patient) were analyzed [17,18]. RT and PR TDRM were identified using the surveillance list in the Stanford University HIV-1 Drug-Resistance Database [15], according to Bennett et al. [14]. This list includes non-polymorphic mutations at 25 RT, and at 18 protease gene positions. Stanford HIVdb algorithm version 9.0 was also used to define HIVdrm within RT, PR and IN. HIVdb returns inferred levels of resistance to 25 antiviral drugs and is regularly updated so that new resistance patterns are included. We used these methods to enable comparison to published national reports; it has already been indicated that there is currently no consolidation, and different methods are in use to determine prevalence data [19]. Subtypes were defined for PR and RT by the REGA HIV-1 subtyping tool version 3.0 [15]. Classification of subtype A sequences into A1 and A6 was achieved using the methods described in [10], and briefly outlined in Appendix A.

### 2.3. Phylogenetic Analysis, Statistical Analysis and Ethical Approval

Partial PR and RT nucleotide sequences were combined using a custom R script following the removal of drug-resistance codons. Phylogenetic analysis of the combined PR and RT sequences was performed separately for each major pure subtype, A, B, or C. Statistical analysis was performed using IBM SPSS statistics 20 version and R studio version 1.2.1335. A detailed description of the phylogenetic and statistical analysis is available in the Appendix A.

The institutional review board of all participating HIV centers approved the study. All data was anonymized, cross-matched, and coded as dual-encoding on demand. A consent waiver was obtained for the study.

## 3. Results

### 3.1. Epidemiology

A total of 3639 PLHIV (Table 1) were diagnosed in Israel in 2010–2018. Main routes of HIV-1 transmission were heterosexual contacts (39.1%), MSM (34.3%) and PWID (10.4%). Only 33.1% of the newly diagnosed HIV-1 were Israeli-born, the rest were immigrants; 24.8% were born in EEU/CA, and 17.6% originated from SSA. The proportion of people infected through heterosexual contacts increased in recent study years (34.8% in 2010–2012; 44.4% in 2016–2018) whereas that of PWID decreased (12.8% in 2010–2012; 6.4% in 2016–2018, *p* < 0.001). On the other hand, the proportion of HIV-1 positive MSM did not change.

Table 2 compares the characteristics of PLHIV for whom country of birth was known (N = 2830). The proportion of transmission groups was different in PLHIV born in Israel and in Western counties (WCEU/NA), who were mostly MSM (78.3% and 82.9%, respectively) compared with other countries were the majority of PLHIV acquired HIV-1 through heterosexual contact. Immigrants from SSA and EEU/CA were diagnosed with lower levels of CD4 (median of 219 cells/mm^3^ and 323 cells/mm^3^, respectively) compared with Israeli born PLHIV (400 cells/mm^3^) or PLHIV originating from WCEU/NA (403 cells/mm^3^) (*p* < 0.001). This pattern was stable within the studied years.

### 3.2. HIV-1 Subtypes and Resistance Mutations

Subtype analysis (available for 1957, 53.4% of the cohort) revealed that the majority (43.9%) were infected with subtype B, and only 17.1% had subtype C (Table 1). Subtype A carriers (25.2%) were further classified into sub-subtypes A6 (22.8%) and A1 (2.4%). Analysis of subtype distribution according to place of birth (Table 2) demonstrated that subtype B was most frequently identified among Israeli born individuals (67.8%), and characterized by infection by MSM, of whom most (83.1%) had subtype B. The A6 carriers originated mainly from EEU/CA (66.9%) and acquired the virus through heterosexual contact (44.4%, 198/446), or injecting drugs (39.5%, 176/446). A1 carriers were mainly (88%) Israeli-born. Most carriers of subtype C were immigrants from SSA (80.2%, *n* = 215, Table 2).

TDRM were observed in 12.1% (of the available 1905 cases) with 6.6, 3.3 and 3.7% of PLHIV carrying NNRTI, NRTI and PI TDRM, respectively. The proportion of TDRM remained stable in 2010–2012, 2013–2015 and 2016–2018 (Table 1). Interestingly, the overall proportion of NNRTI TDRM was higher in Israeli-born individuals (8.6%) compared with those born elsewhere (3.3–7.2%, Table 2). When sequences were analyzed using the Stanford HIVdb, a higher prevalence of overall HIVdrm (21.7%) was observed, with 12.5, 6.2 and 5.9% NNRTI, NRTI and PI-resistance mutations, respectively.

Table 3 lists the number and proportion of the most prominent resistance mutations. L90M PI mutation (in 2%, 39/1905) was significantly more frequent in subtype B. The NRTI mutation A62V, (2.7%, 51/1905) was significantly more prominent in sub-subtype A6. M184IV NRTI TDRM (0.9%, 18/1905), was identified in all major subtypes. Three NNRTI mutations were highly prevalent: K103N (in 4.5%, 86/1905), E138A (in 3.7%, 70/1905) and E138Q (in 1.4%, 26/1905). Both K103N and E138Q (35%, 16/46 and 48%, 22/46, respectively) were highly abundant in sub-subtype A1 sequences (*p* < 0.001). Moreover, K103N/E138Q or K103S/E138Q combination was observed in 32.6% (15/46) and 13% (6/46) of A1 carriers, respectively. No major IN mutations were identified; however, the integrase resistance associated accessory mutation L74I found in 9.2% (44/479) of the patients was more prominent in A6 (38/112, 34%), compared with other subtypes, and also found to be highly prevalent (80%) in PLHIV from EEU/CA. Results of all resistant mutations were detailed in Appendix A.

### 3.3. Multidimensional Mapping

To further assess the probability of having TDRM, multivariate logistic regression was performed. Probability of TDRM (0.0–1.0) was visualized by multidimensional mapping (Figure 1). The highest probability of having TDRM (≥0.5) was observed for Israeli-born individuals below the age of 50 (OR: 2.07, 1.50–2.86 of 95% CI, *p* < 0.001) carrying HIV-1 A1 sub-subtype (OR: 13.43, 6.57–27.52 of 95% CI, *p* < 0.001 for male and OR: 13.84, 2.98–64.19 of 95% CI, *p* = 0.001 for female). Details and results of this analysis are presented in Appendix A.

### 3.4. Phylogenetic Analysis

Phylogenetic analysis of the most prevalent subtypes (A, B and C) revealed unexpected clusters of PLHIV, some also sharing common drug-resistance mutations. For instance, among subtype C (*n* = 313), which is typical for immigrants from SSA, a cluster of individuals from a northern district of Israel was observed, of which 68.8% (11/21) were Israeli-born MSM. Clusters of individuals originating from EEU/CA with those from IL, mostly infected through heterosexual contacts, could also be observed (Figure 2, Appendix A).

Subtype B (N = 740, Figure 3), comprising mainly MSM, could be divided into 86 clusters of which five were large, and included dozens of individuals. One such cluster was characterized by the K103N mutation in more than a half of its members (Appendix A, Cluster B4). Another included mainly carriers of the PR L90M (92.6%, 25/27, Appendix A, Cluster B5).

Subtype A (N = 482, Figure 4) analysis revealed a tight A1 cluster including mainly Israeli-born individuals (79%, 34/43) 59% of whom (20/34) were MSM. Many of all A1 (46%) had a combined NNRTI (K103N/S, E138Q) resistance mutation. Among A6, a chain of transmission of 68 individuals was observed. Most (91.2%, 62/68) were PWID (Appendix A).

## 4. Discussion

This study shows a change in the characteristics of newly diagnosed HIV-1 infected individuals in 2010–2018. While, as a result of past waves of immigration from Ethiopia [20,21], HIV-1 positive immigrants from SSA dominated the local epidemic before 2010, in the years 2010–2018, only 17.6% were immigrants from SSA. In addition, the proportion of MSM (34.3%) increased, similar to the trend observed during these years in Europe where MSM accounted for 40% of all new HIV-1 diagnoses [22]. Overall, more than half (65%) of newly identified PLHIV were immigrants, of which a quarter (24.8%, *n* = 901) were immigrants from EEU/CA. These results define a completely new landscape of individuals diagnosed with HIV-1 in Israel.

The change in these proportions could also be corroborated by the overall distribution of the major HIV-1 subtypes. Previously, subtype C characterizing the SSA epidemic, was the most frequent subtype followed by subtype B (infecting mainly MSM) and subtype A related to PWID [6,7]. Here, the most prevalent subtype was subtype B (43.9%) and only 17.1% had subtype C. Subtype A carriers, could now be divided into A1 and A6. Most PWID were found to be infected with sub-subtype A6, which is prevalent in Russia [10], while sub-subtype A1 was mainly identified in Israeli-born MSM. Recent reports have shown that, in Europe, immigration also affects the current epidemic, which is evolving with an increase in the prevalence of non-B subtypes [23].

Intermingling between the traditional transmission groups, a phenomenon not noted in previous years, was demonstrated by the phylogenetic analysis. For example, Israeli-born MSM within a subtype C cluster, and PLHIV born in places other than SSA clustered with subtype C sequences. Moreover, some unusual cases of SSA-born carriers with subtype A6 were identified. These results mainly suggest that local transmission is ongoing.

It was already reported that PLHIV identified in 2011–2015 [24] were diagnosed late, with median CD4 counts below CD4 ≤ 350 cells/mm^3^ [25]. Similar findings were found here, in more recent years. Particularly, low initial CD4 counts characterized PLHIV born in SSA, and EEU/CA (219 and 323 cells/mm^3^ respectively) compared with PLHIV originating in Israel or in WCEU/NA (400 and 403 cells/mm^3^, respectively). Late diagnosis of HIV remains a major challenge in the HIV epidemic globally. In Europe about 50% of all people living with HIV are diagnosed late, after infection has occurred [26]. Methods that could increase the rates of early HIV-1 diagnosis, such as increasing the awareness of HIV transmission routes, communicating the availability of free-of-charge diagnostic tests, and the benefits of early testing and early entry to HIV care, should be developed. 

TDRM were identified in 12.1% and HIVdrm (which counts mutations in the polymorphic RT A62 and E138 sites) in >20% of the population. Currently, there are differences between countries with respect to methodology of analyzing baseline resistance mutations [19]. To enable better comparison to published prevalence data, we analyzed TDRM according to the surveillance SDRM list, as well as HIVdrm (including all mutations affecting current therapies) according to the Stanford HIVdr algorithm. In a recent report that used the Stanford HIVdr for analysis, the overall proportion of resistance mutations was 13.5% in nine European countries in 2017, a much lower proportion than that identified herein in Israel [19]. In their 2017 report, the WHO announced that countries with a high rate (>10%) of HIV-1 drug-resistance mutations, especially those where the use of non-NNRTI in first-line treatment regimen is not optional, may consider using pretreatment drug-resistance to guide first-line treatment [27]. In Israel, integrase inhibitors in first-line therapy were introduced during 2014. However, currently, baseline resistance testing is still recommended and, with the ongoing local transmission, is required to enable continuous surveillance of this disease.

The most often detected mutation was K103NS. It was significantly more prominent in sub-subtype A1 (48%, 22/46), but also in subtype B (6.4%, 54/841), where active networks were observed, suggesting the risk of ongoing transmission of such resistant viruses [2,5]. Mutations at the polymorphic position E138 that affect Etravirine and rilpivirine, were the most prominent NNRTI mutations (5.7%), most often detected in sub-subtype A1 (54%, 25/46) compared with all other subtypes (*p* < 0.001). Moreover, clusters of sub-subtype A1 with double mutations—either K103N and E138Q or K103S and E138Q- in the same individuals—were observed. Such clusters of multiple NNRTI resistance mutations, especially the more advanced K103S with E138Q combinations [15], may suggest transmission of viruses from individuals treated with historical NNRTIs who were also exposed to rilpivirine-containing regimens [28]. Circulation of these double mutants is probably ongoing, as they were observed in the cluster of A1: Israeli-born, aged < 50 years.

The most prominent NRTI resistance mutation was A62V, prevalent especially in PLHIV with sub-subtype A6 originating from Eastern Europe [10]. Since A62V does not interfere with current regimens, it is not clinically significant. M184IV, related to high-level resistance to lamivudine (3TC) and emtricitabine (FTC), was the only clinically relevant NRTI mutation identified. Since pre-exposure prophylaxis therapy is based also on 3TC [29], M184V should still be monitored. The L90M PI mutation, which was identified mainly in subtype B (4%), is clinically irrelevant as nelfinavir is not in use anymore.

Although no integrase strand transfer inhibitor (INSTI) major mutations were found, the low-genetic-barrier polymorphic L74I mutation was prevalent (mainly in A6). As the potency of several INSTI could be affected if mutations like G140 and Q148 are combined with the L74-mutated site [30], this mutation is of relevance to the clinician.

The majority of PLHIV with resistance mutations were born in Israel. A previous study conducted in Israel between 1999 and 2003 identified resistance mutations in 14.8% of newly diagnosed naïve PLHIV, 28.6% of whom were known to have been infected in Israel [11]. Indeed, here, multidimensional mapping demonstrated that the largest probability of having TDRM (0.5, 0.0–1.0) is among young (<50 years) Israeli-born PLHIV. Therefore, the importance of early HIV-1 diagnosis in Israeli-born individuals could not be underestimated.

Our study had several inherent limitations. First, resistance testing was performed in only 52.3% of the newly diagnosed population. This is an inherent limitation that needs to be further explored. Second, a reporting bias may be present, such as sexual behavior, drug injection or previous treatment. To partially overcome this obstacle, data was compared and cross-matched before the analysis. Samples from patients born elsewhere, with no clear treatment history, were excluded. Despite these limitations, the availability of demographic data on all naïve PLHIV identified in Israel in 2010–2018 were allowed to specifically study their contribution to the local epidemic.

In summary, the epidemiology of HIV-1 positive immigrants is changing, shifting to individuals born in EEU/CA, infected through heterosexual contact with a reduction in the proportion of newly identified patients from SSA. The high proportion of Israeli-born treatment naïve PLHIV with resistance mutations, and the probability of local HIV-1 transmission, suggests that resistance testing at baseline should continue. Standardized drug-resistance interpretation and surveillance using an identical algorithm will enable a more robust comparison of transmitted drug-resistance between countries. Unexpected clusters of PLHIV infected with viral subtypes which are uncommon for their transmission groups also emphasize this need. Expanded testing to all target populations should be offered to increase the rates of early HIV-1 diagnosis.

## Figures and Tables

**Figure 1 viruses-14-00071-f001:**
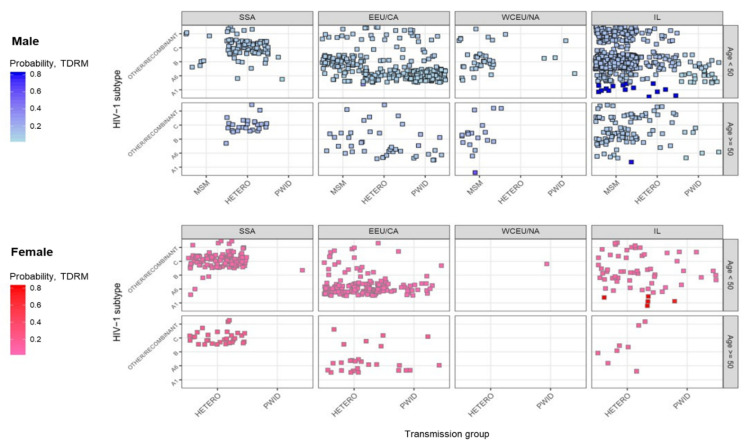
Probability of TDRM by sex, age, birthplace, HIV-1 subtype and transmission group. Following logistic regression analysis assessing predictors of TDRM (Appendix A), all predicted values of mean of response related to TDRM were preserved. Subsequently, the probability of TDRM (0.0–1.0) was derived. Here, multi-dimensional mapping visualizes the results of this analysis. TDRM-transmitted drug-resistance mutations; SSA—Sub-Saharan Africa; EEU/CA—Eastern Europe and Central Asia; WCEU/NA—Western and Central Europe and North America; IL—Israel; PWID—people who use injection drug; MSM—men who have sex with men; HETERO—heterosexual contacts.

**Figure 2 viruses-14-00071-f002:**
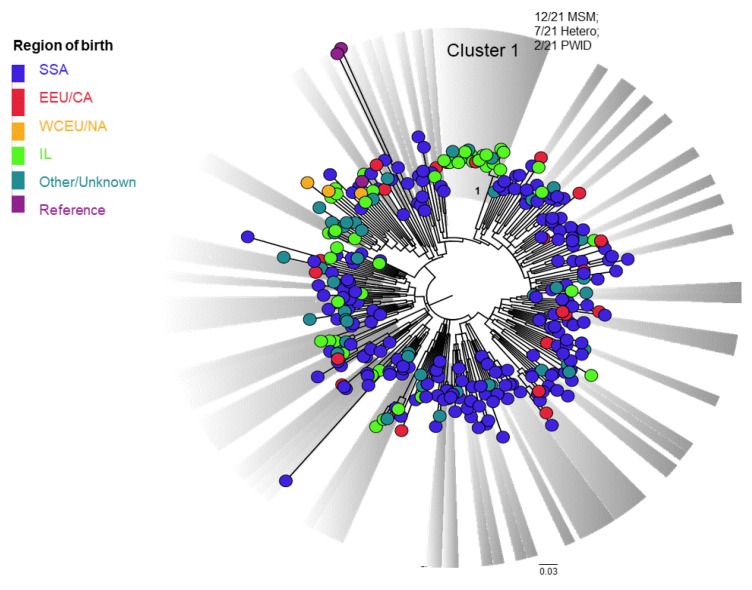
Phylogenetic analysis of subtype C sequences, N = 313. A phylogenetic tree of the pol region of subtype C sequences from 313 HIV-1 patients using the maximum likelihood (ML) with GTR + G + I model is shown. Included are three reference sequences (Appendix A). The tree was visualized in Fig Tree version 1.4.4. Clusters (bootstrap values > 0.7) are colored in grey. The regions of origin are colored as follows: SSA (Sub-Saharan Africa) in navy blue; EEU/CA (Eastern Europe and Central Asia) in red; WCEU/NA (Western and Central Europe and North America) in orange; IL (Israel) in light green; other/unknown birthplace in turquoise; reference sequences in purple. PWID—people who use injection drug; MSM—men who have sex with men; HETERO—heterosexual contacts.

**Figure 3 viruses-14-00071-f003:**
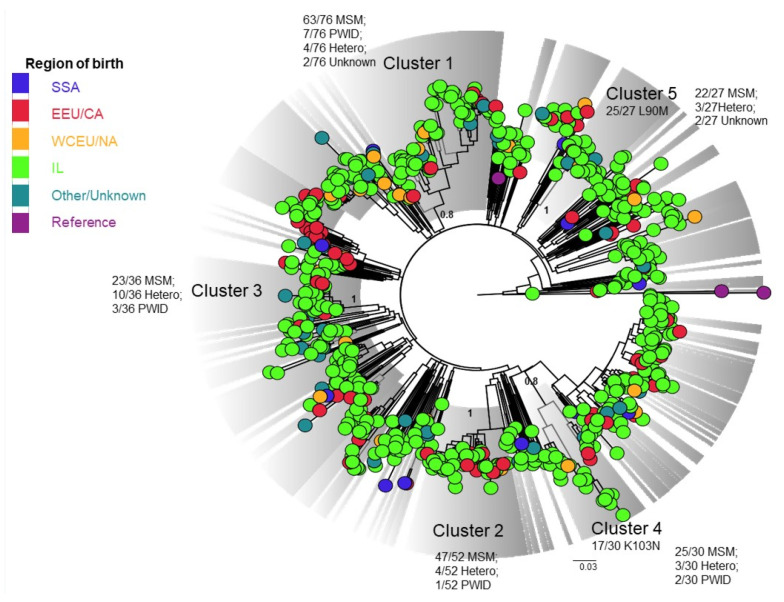
Phylogenetic analysis of subtype B sequences, N = 740. A phylogenetic tree of the pol region of subtype B sequences from 740 HIV-1 patients using the maximum likelihood (ML) with GTR + G + I model is shown. Included are three reference sequences (Appendix A). The tree was visualized in Fig Tree version 1.4.4. Clusters (bootstrap values > 0.7) are colored in grey. Countries of origin are colored as follows: SSA (Sub-Saharan Africa) in navy blue; EEU/CA (Eastern Europe and Central Asia) in red; WCEU/NA (Western and Central Europe and North America) in orange; IL (Israel) in light green; other/unknown birthplace in turquoise; reference sequences in purple. PWID—people who use injection drug; MSM—men who have sex with men; HETERO—heterosexual contacts.

**Figure 4 viruses-14-00071-f004:**
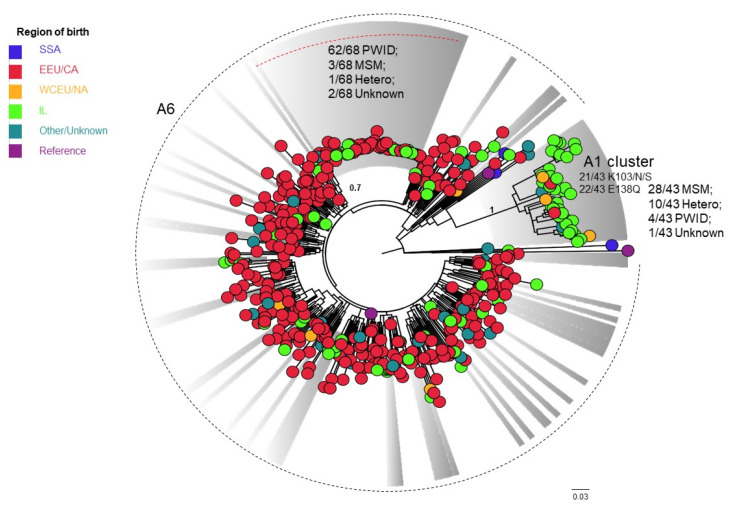
Phylogenetic analysis of subtype A sequences, N = 482. A phylogenetic tree of the pol region of subtype A sequences from 482 HIV-1 patients using the maximum likelihood (ML) with GTR + G + I model is shown. Included are three reference sequences (Appendix A). The tree was visualized in Fig Tree version 1.4.4. Clusters (bootstrap values > 0.7) are colored in grey. Countries of origin are colored as follows: SSA (Sub-Saharan Africa) in navy blue; EEU/CA (Eastern Europe and Central Asia) in red; WCEU/NA (Western and Central Europe and North America) in orange; IL (Israel) in light green; other/unknown birthplace in turquoise; reference sequences in purple. The dashed line presents sub-subtype A6. PWID—people who use injection drug; MSM—men who have sex with men.

**Table 1 viruses-14-00071-t001:** Characteristics of newly diagnosed PLHIV, by years of diagnosis.

	All (N = 3639)	2010–2012 (N = 1264)	2013–2015 (N = 1294)	2016–2018 (N = 1081)	*p*-Value
Age at diagnosis (years), median (IQR) (*n* = 3500)	37 (30–45)	36 (29–44)	37 (31–45)	38 (32–46)	<0.001
Sex, *n* (%)					
Female	1032 (28.4)	361 (28.6)	355 (27.4)	316(29.2)	0.025
Male	2561 (70.4)	883 (69.9)	922 (71.3)	756 (69.9)	
Trans people	12 (0.3)	1 (0.1)	5 (0.4)	6 (0.6)	
Unknown	34 (0.9)	19 (1.5)	12 (0.9)	3 (0.3)	
Place of birth, *n* (%)					
Sub-Saharan Africa	640 (17.6)	267 (21.1)	191 (14.8)	182 (16.8)	<0.001
Eastern Europe and Central Asia	901 (24.8)	266 (21)	334 (25.8)	301 (27.8)	
Israel	1206 (33.1)	415 (32.8)	450 (34.8)	341 (31.5)	
West/Central Europe/North America	83 (2.3)	23 (1.8)	28 (2.2)	32 (3)	
Other/Unknown	809 (22.2)	293 (23.2)	291 (22.5)	225 (20.8)	
Transmission Groups, *n* (%)					
Men who have sex with men	1248 (34.3)	432 (34.2)	451 (34.9)	365 (33.8)	<0.001
Heterosexual contacts	1424 (39.1)	440 (34.8)	504 (38.9)	480 (44.4)	
Injecting drug users	380 (10.4)	162 (12.8)	149 (11.5)	69 (6.4)	
Unknown	587 (16.1)	230 (18.2)	190 (14.7)	167 (15.4)	
CD4 (cells/mm^3^), N (%)	N = 1899	N = 704 (37.1)	N = 730 (38.4)	N = 465 (24.5)	
CD4 (cells/mm^3^), median (IQR)	354 (187–544)	351 (183–540)	357 (193–544)	360 (186–557)	0.91
HIV-1-RNA (Log copies/mL), N (%)	N = 2103	N = 771 (36.7)	N = 814 (38.7)	N = 518 (24.6)	
HIV-1-RNA (Log copies/mL), median (IQR)	4.7 (4.1–4.7)	4.7 (4.0–5.2)	4.7 (4.1–5.3)	4.9 (4.3–5.5)	<0.001
HIV-1 Subtype (*n* = 1957), N (%)	N = 1957	N = 694 (35.5)	N = 741 (37.9)	N = 522 (26.7)	
A1	46 (2.4)	15 (2.2)	24 (3.2)	7 (1.3)	<0.001
A6	446 (22.8)	156 (22.5)	167 (22.5)	123 (23.6)	
B	860 (43.9)	336 (48.4)	330 (44.5)	194 (37.2)	
C	335 (17.1)	130 (18.7)	119 (16.1)	86 (16.5)	
Other/Recombinant form	270 (13.8)	57 (8.2)	101 (13.6)	112 (21.5)	
TDRM by class (N = 1905), *n* (%)	231 (12.1)	70 (10.5)	96 (13.3)	65 (12.6)	0.221
NNRTI	125 (6.6)	39 (5.8)	48 (6.6)	38 (7.4)	0.282
NRTI	62 (3.3)	28 (4.2)	19 (2.6)	15 (2.9)	0.188
PI	71 (3.7)	19 (2.8)	30 (4.1)	22 (4.3)	0.177
HIVdrm by class (N = 1905), *n* (%) ECDC algorithm	413 (21.7)	122 (18.3)	168 (23.2)	123 (23.9)	0.015
NNRTI	238 (12.5)	72 (10.8)	105 (14.5)	61 (11.9)	0.466
NRTI	118 (6.2)	42 (6.3)	35 (4.8)	41 (8)	0.3
PI	112 (5.9)	28 (4.2)	50 (6.9)	34 (6.6)	0.061

Data are presented as *n* (%) or median (IQR); IQR, interquartile range; significance for differences was measured using Chi-squared test, Fisher’s Exact test, or Kruskal–Wallis test. TDRM—transmitted drug-resistance mutations; HIVdrm—HIV-1 drug-resistance mutations; ECDC—European Centre for Disease Prevention and Control; NNRTI—non-nucleoside reverse transcriptase inhibitors; NRTI—nucleotide reverse transcriptase inhibitors; PI—protease inhibitors.

**Table 2 viruses-14-00071-t002:** Characteristics of newly diagnosed PLHIV (2010–2018) with known * place of birth (*n* = 2830).

	Sub-Saharan Africa (N = 640)	Eastern Europe/Central Asia (N = 901)	Israel (N = 1206)	Western/Central Europe/North America (N = 83)	*p*-Value
Age at diagnosis, N = 2784, (98.4%)	605 (21.7)	892 (32)	1205 (43.3)	82 (2.9)	
Years, Median (IQR)	39 (32–48)	38 (32–45)	35 (28–43)	43 (31–52)	<0.001
≥50, *n*= 426 (15.3%)	129 (21.3)	129 (14.1)	146 (12.1)	25(30.5)	<0.001
<50, *n* = 2358 (84.7%)	476 (78.7)	773 (85.9)	1059 (87.9)	57(69.5)	
Sex, N = 2819 (99%)	631(22.4)	900 (31.9)	1205 (42.7)	83 (2.9)	
Female, *n* = 775 (27.5%)	327 (51.8)	352 (39.1)	92 (7.6)	4 (4.8)	<0.001
Male, *n* = 2033 (72.1%)	304 (48.2)	545 (60.6)	1106 (91.8)	78 (94)	
Trans people, *n* = 11 (0.4%)	0	3 (0.3)	7 (0.6)	1 (1.2)	
Transmission Groups, (N = 2792), *n* (99%)	640 (22.9)	901 (32.3)	1175 (42.1)	76(2.7)	
Men who have sex with men, *n* = 1122 (40.2%)	11 (1.7)	128 (14.2)	920 (78.3)	63 (82.9)	<0.001
Heterosexual contacts, *n* = 1328 (47.6%)	625 (97.7)	513 (56.9)	182 (15.5)	8 (10.5)	
Injecting drug users, *n* = 342 (12.2%)	4 (0.6)	260 (28.9)	73 (6.2)	5 (6.6)	
HIV-1 subtype (*n* = 1787), *n* (91.6%)	268 (15)	490 (27.4)	972 (54.4)	57 (3.2)	
A1, *n* = 42 (2.4%)	0	2 (0.4)	37 (3.8)	3 (5.3)	<0.001
A6, *n* = 412 (23.1%)	4 (1.5)	328 (66.9)	75 (7.7)	5 (8.8)	
B, *n* = 809 (45.3%)	16 (6)	99 (20.2)	659 (67.8)	35 (61.4)	
C, *n* = 290 (16.2%)	215 (80.2)	23 (4.7)	48 (4.9)	4(7)	
Other/recombinant, *n* = 234 (13.1%)	33 (12.3)	38 (7.8)	153 (15.7)	10 (17.5)	
CD4 (N = 1722), *n* (61%)	(N = 254)	(N = 469)	(N = 953)	(N = 46)	
CD4 (cells/mm^3^) Median (IQR)	219 (103–369)	323 (141–540)	400 (242–588)	403 (267–552)	<0.001
HIV-1 RNA, N = 1895, *n* (67.6%)	(N = 303)	(N = 525)	(N = 1010)	(N = 57)	
HIV-1 RNA log copies/mL, Median (IQR)	4.7 (3.8–5.5)	4.7 (4.1–5.3)	4.8 (4.2–5.3)	4.8 (4.1–5.2)	0.132
TDRM by class, N = 1741, *n* (%)	32 (12.8)	39 (8.1)	140 (14.7)	9 (16.4)	0.004
NNRTI	18 (7.2)	16 (3.3)	82 (8.6)	3 (5.5)	0.003
NRTI	14 (5.6)	18 (3.7)	23 (2.4)	3 (5.5)	0.057
PI	8 (3.2)	12 (2.5)	44 (4.6)	4(7.3)	0.120
HIVdrm by class, N = 1741, *n* (%) ECDC algorithm	48 (19.2)	114 (23.7)	205 (21.5)	17 (30.9)	0.205
NNRTI	35 (14)	50 (10.4)	128 (13.4)	7 (12.7)	0.364
NRTI	16 (6.4)	54 (11.2)	30 (3.1)	6 (10.9)	<0.001
PI	10 (4)	25 (5.2)	64 (6.7)	8 (14.5)	0.018

Data are presented as *n* (%) or median (IQR); IQR, interquartile range; significance for differences was measured using Chi-squared test, Fisher’s Exact test, or Kruskal–Wallis test. TDRM—transmitted drug-resistance mutations; HIVdrm—HIV-1 drug-resistance mutations; ECDC—European Centre for Disease Prevention and Control; NNRTI—non-nucleoside reverse transcriptase inhibitors; NRTI—nucleotide reverse transcriptase inhibitors; PI—protease inhibitors. * PLHIV with unknown place of birth were removed from this analysis.

**Table 3 viruses-14-00071-t003:** Proportion of most frequently and/or clinically relevant detected resistance mutations in newly diagnosed people living with HIV-1 (PLHIV) in Israel 2010–2018.

Drug Class	Mutation	All, N = 1905	A1, N = 46	A6, N = 436	B, N = 841	C, N = 318	Other, N = 264	*p*-Value
PI, *n* (%)	L90M	39 (2)		2 (0.4)	34 (4)	1 (0.3)	2 (0.8)	B vs. A6/C, <0.001
NRTI, *n* (%)	A62V	51 (2.7)		49 (11)		2 (0.6)		A6 vs. C, <0.001
	M184IV	18 (0.9)		7 (1.6)	5 (0.6)	6(1.9)		B vs. C, 0.043
NNRTI, *n* (%)	K103N	86 (4.5)	16 (35)	5 (1.1)	53 (6.3)	7 (2.2)	5 (1.9)	A1 vs. all, <0.001 A1 vs. B/Other, <0.001 A6/B/C vs. Other ≤ 0.01 A1 vs. B, <0.001 NS A1 vs. A6/B/C, <0.001
	K103S	8 (0.4)	6 (13)		1 (0.1)		1 (0.4)	
	E138A	70 (3.7)	1 (2.2)	17 (3.9)	21 (2.5)	10 (3.1)	21 (8)	
	E138G	10 (0.5)	2 (4.3)	4 (0.9)	1 (0.1)	2 (0.6)	1 (0.7)	
	E138K	3 (0.2)		1 (0.2)	1 (0.1)	1 (0.3)		
	E138Q	26 (1.4)	22 (48)	1 (0.2)	2 (0.2)		1 (8)	
INSTI, *n* (%)		All, N = 479	A1, N = 8	A6, N = 112	B, N = 202	C, N = 65	Other, N = 92	
	L74I	44 (9.2)		38 (34)	3 (1.5)	1 (1.5)	2 (2.2)	A6 vs. all, <0.001

Data are presented as *n* (%). Differences in proportions were measured using the chi-squared test. PI—protease inhibitors; NRTI—nucleotide reverse transcriptase inhibitors; NNRTI—non-nucleoside reverse transcriptase inhibitors; INSTI—integrase strand transfer inhibitors; NS—non-significant.

## Data Availability

Raw data can be provided upon request from the authors.

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
