# Peer review of "Epidemiology and Transmitted HIV-1 Drug Resistance among Treatment-Naïve Individuals in Israel, 2010–2018"

_viruses, 2021, doi:10.3390/v14010071_

Round 1

Reviewer 1 Report

The authors present the relatively recent characteristics of HIV epidemic in Israel. They show demographic and clinical data for 3639 newly diagnosed individuals in 2010-2018, combined with sequencing data for over half of them. The results show changes in the characteristics, generally caused by the changes of the ethnicity of immigrants entering Israel. The relatively high prevalence of transmitted drug resistance among the local population confirms the need for routine baseline testing and identification of key population most vulnerable for infection. Furthermore, the results indicate there is an ongoing transmission cluster of resistant viruses that should be a concern.   

Minor comments

Abstract:

Line 40-41: It is not clear what the stated proportion of SSA and EEU/CA refers to, since it is mentioned that the proportions changed in the 2010-2018.

Line 45-48: The relationship within the subtypes and transmission groups is not presented clearly. The authors should first state the obtained results (for example: “subtype B was present in 45.3%, subtype A in 25.5% and subtype C in 16.2% of individuals”) and then go into the subtype distribution among transmission routes or subtype representation in specific transmission groups, such as MSM. After this, the further analysis of phylogenetic relations within subtypes can be reported.

Line 47-48: The authors state: “Israelis, <50 years old, carrying A1 had the highest risk for having TDRM”. Was this finding significant? Opposed to whom - what was the reference group here, immigrants, older people, other subtypes?  

Introduction:

Line 56-63: The authors state: “Before 2010, the Israeli Ministry of Health (MoH) reported 6579 HIV-1 cases, of whom 41.3% originated from countries with generalized epidemics in Sub-Saharan Africa, SSA, (mostly by heterosexual transmission), 21.3% were men who have sex with men (MSM), 13.4% were people who use injection drugs (PWID) and 12.1 % were heterosexuals who were not from countries in SSA and were not PWID (1).” The sentence should be rephrased, since this now reads that among 41.3% from SSA 21.3% were MSM, etc. This was not the case and should be rephrased to state that 41.3% originated from SSA and in a separate sentence state the transmission routes of HIV-1, such as MSM, PWID, heterosexual, etc. Also, the statement that 12.1% were heterosexuals who were not from SSA and were not PWID is confusing. The overall proportion of heterosexual transmission is not clear.

Line 65: Please refer to HIV-1 subtypes and not genotypes.

Line 67-70: The authors state: “Recently, a new classification of sub-subtype A into A1 and A6 has been reported. A1 sub-subtype is predominantly found in SSA and A6 is unique to regions in the  EEU.” This information is not recent. Recently, a new classification of subtype A was proposed with subtype A subdivided into six sub-subtypes (A1–A4, A6, A7) (Désiré, et al., Retrovirology, 2018). Furthermore a new sub-subtype was detected, sub-subtype A8 (Mendes, et al., Viruses, 2021).

Line 97-98: Please do not report units of measurement (log copies/ml, cells/mm3) when stating clinical data, or at least put them in brackets.

Line 99-100: EEU/CA abbreviation has already been introduced in the text, therefore only use the abbreviation.

Results:

Line 133-134: The authors state: “The proportion of people infected through heterosexual contacts (44.4%) increased in recent study years (2016-2018) while that of PWID (6.4%) decreased (p<0.001). On the other hand, the proportion of HIV-1 positive MSM did not change.” It is not clear what the stated proportions represent. Please indicate what is this proportion compared to and the respective time frame.

Table 1: The group “Other” under the characteristic “Place of birth” should probably be renamed “Other/unknown”, since Table 2 reveals that this information was available for only 2830 PLHIV.

Table 2: The authors divide the patients into two age groups at diagnosis (≥50 and <50). What was the rationale behind this? Usually, more age groups are analyzed. The problem is that among the four groups analyzed according to the place of birth, in three groups the IQR is below 50, which makes for even less sense of analyzing the data in this way.

Table 2: The sum of PLHIV in the row “Sex, N=2819 (99%)” is not correct, it should be 631, 900, 1205, and 83. Proportions in the brackets should be corrected, accordingly.

Table 2: The sum of “Transmission groups” is not 2796 but 2792, please correct.

Table 2: In the row “A6” in the last column please correct the values, it should be “5(8.8)” and not “8.8(5)”.

Line 170-172: The authors state: “Interestingly, the overall proportion of any TDRM and especially of NNRTI TDRM was higher in Israeli-born individuals (8.6%) compared to those born elsewhere (3.3-7.2%, Table 2).” This statement is not correct, since overall proportion of TDRM was the highest in WCEU/NA (16.4%).

Line 179-182: The authors state: “Two NNRTI mutations were highly prevalent: K103N/S (in 4.9%, 94/1905) and the E138AGKQ (in 5.7%, 109/1905). Both K103N/S and E138AGKQ (48 %, 22/46 and 54%, 25/46, respectively) were highly abundant in sub-subtype A1 sequences (p<0.001).” Please refer to mutations separately, since K103N/S is not a single mutation, it is two mutations, K103N and K103S. The same is true for E138AGKQ, these are four different mutations with four different mutation prevalences and also the impact of these mutations on the resistance to drugs can be different.

Table 3 and Supplementary Table S4: Please list all the detected mutations and not only per site (for example K103N and K103S separately, and not K103NS). Please correct this also in the text. The authors chose to analyze if the mutations are significantly related to the subtypes and this might have biased their results. Some mutations are known to be more frequent in some subtypes.

Line 197-201: The authors state: “The highest probability of having TDRM (>0.5) was observed for Israeli - born individuals below the age of 50 (OR: 2.07, 1.50-2.86 of 95% CI, p<0.001) carrying HIV-1 A1 sub-subtype (OR: 13.43, 6.57-27.52 of 95% CI, p<0.001 for male and OR: 13.84, 2.98-64.19 of 95% CI, p=0.001 for female).” This is an interesting finding, did you also check if there is any correlation with transmission risk, specifically MSM, and CD4 cell count (low vs. high; – can be somewhat an indicator of recent vs. long-term infection)?

Discussion:

Line 232: The authors state: “Overall, more than (65%) of newly identified PLHIV were immigrants,” Please correct to “more than half (65%)”.

Line 292: The authors mention PrEP. It would be interesting to know how is the situation regarding PrEP in Israel? Can it be prescribed, is it reimbursed?

Line 306-307: Please explain why only 52.3% had a resistance test performed, since you state that in Israel resistance testing is recommended at baseline.

Line 314-316: The authors state: “In summary, the epidemiology HIV-1 in Israel is changing, shifting to individuals born in EEU/CA infected through heterosexual contacts with a reduction in the proportion of newly identified patients from SSA.” This is not correct, since the majority of individuals in this study were born in Israel and not EEU/CA. This change to EEU/CA is however seen in HIV-positive immigrants and not in the overall HIV-positive population.

Figures:

Figures S1-3: The authors list the mutations on some of the clusters. It is however not shown, how prevalent are these mutations in the clusters and how are the detected mutations observed elsewhere, outside the highlighted clusters. This information is also lacking in the caption of these figures. What mutations were inspected? Do the authors limit themselves only on TDRM? Inclusion of more data on this topic would be interesting to see, to better observe the spread of certain TDRM in the population. The authors could consider adding an additional table to highlight the largest transmission clusters and the relevant mutation observed there, together with the demographic data. In addition, seroconversion data is presented in some of the clusters in the figures. There is no mentioning of this trait in the text. If the authors wish to include this, it should be mentioned in the Method section and in the caption of the figures. In this regard, if the data of seroconversion is included, it could have been also analyzed to see if there is any correlation between seroconversion and the presence of TDRM, etc.

Author Response

Dear Reviewer

Thank you for your very thorough review. I really think your remarks have improved our manuscript.

Below please find our point by point reply to your remarks

Abstract:

Line 40-41: It is not clear what the stated proportion of SSA and EEU/CA refers to, since it is mentioned that the proportions changed in the 2010-2018.

Reply: Thank you. We corrected to:

The proportion of individuals from SSA decreased through the years 2010-2018 (21.1% in 2010-2012;  16.8% in 2016-2018) while that of those from EEU/CA increased significantly (21% in 2010-2012; 27.8% in 2016-2018, p<0.001)

Line 45-48: The relationship within the subtypes and transmission groups is not presented clearly. The authors should first state the obtained results (for example: “subtype B was present in 45.3%, subtype A in 25.5% and subtype C in 16.2% of individuals”) and then go into the subtype distribution among transmission routes or subtype representation in specific transmission groups, such as MSM. After this, the further analysis of phylogenetic relations within subtypes can be reported.

Reply: corrected

" Subtype B was present in 43.9%, subtype A in 25.2% (A6 in 22.8 and A1 in 2.4%) and subtype C in 17.1% of individuals"

Line 47-48: The authors state: “Israelis, <50 years old, carrying A1 had the highest risk for having TDRM”. Was this finding significant? Opposed to whom - what was the reference group here, immigrants, older people, other subtypes?

Reply: This was a significant finding. As detailed in the supplementary file we preformed multivariate logistic regression analysis to test factors associated with TDRM by drug class. All statistically significant univariate predictors of TDRM (sex, age at diagnosis (<50, >50), country of birth (SSA, EEU/CA, Israel, WCEU/NA), risk group (MSM, heterosexual contacts, PWID) and HIV-1 subtypes (B, A1, A6, C, other/recombinants) were considered in multivariate analysis. TDRM probabilities (0.0-1.0) were than calculated and the highest risk was for this group of patients.

Introduction:

Line 56-63: The authors state: “Before 2010, the Israeli Ministry of Health (MoH) reported 6579 HIV-1 cases, of whom 41.3% originated from countries with generalized epidemics in Sub-Saharan Africa, SSA, (mostly by heterosexual transmission), 21.3% were men who have sex with men (MSM), 13.4% were people who use injection drugs (PWID) and 12.1 % were heterosexuals who were not from countries in SSA and were not PWID (1).” The sentence should be rephrased, since this now reads that among 41.3% from SSA 21.3% were MSM, etc. This was not the case and should be rephrased to state that 41.3% originated from SSA and in a separate sentence state the transmission routes of HIV-1, such as MSM, PWID, heterosexual, etc. Also, the statement that 12.1% were heterosexuals who were not from SSA and were not PWID is confusing. The overall proportion of heterosexual transmission is not clear.

Reply: Thank you for this important remark

We revised as follows:

Before 2010, the Israeli Ministry of Health (MoH) reported 6579 HIV-1 cases, of whom 41.3% originated from countries with generalized epidemics in Sub-Saharan Africa, SSA. The transmission groups of the 6579 HIV-1 cases were men who have sex with men (MSM), 21.3% and people who use injection drugs (PWID), 13.4%. While most individuals from SSA were considered to be infected though heterosexual transmission, only 12.1 % of those originating from countries other than SSA were considered to be infected through heterosexual transmission (1).

Line 65: Please refer to HIV-1 subtypes and not genotypes.

Reply: corrected

Line 67-70: The authors state: “Recently, a new classification of sub-subtype A into A1 and A6 has been reported. A1 sub-subtype is predominantly found in SSA and A6 is unique to regions in the  EEU.” This information is not recent. Recently, a new classification of subtype A was proposed with subtype A subdivided into six sub-subtypes (A1–A4, A6, A7) (Désiré, et al., Retrovirology, 2018). Furthermore a new sub-subtype was detected, sub-subtype A8 (Mendes, et al., Viruses, 2021).

Reply: Thanks, we corrected accordingly

Line 97-98: Please do not report units of measurement (log copies/ml, cells/mm3) when stating clinical data, or at least put them in brackets.

Reply: Thanks, removed

Line 99-100: EEU/CA abbreviation has already been introduced in the text, therefore only use the abbreviation.

Reply: Thanks, removed

Results:

Line 133-134: The authors state: “The proportion of people infected through heterosexual contacts (44.4%) increased in recent study years (2016-2018) while that of PWID (6.4%) decreased (p<0.001). On the other hand, the proportion of HIV-1 positive MSM did not change.” It is not clear what the stated proportions represent. Please indicate what is this proportion compared to and the respective time frame.

Reply: Corrected as follows:

The proportion of people infected through heterosexual contacts increased in recent study years (34.8% in 2010-2012; 44.4% in 2016-2018) while that of PWID decreased (12.8% in 2010-2012; 6.4% in 2016-2018, p<0.001).

Table 1: The group “Other” under the characteristic “Place of birth” should probably be renamed “Other/unknown”, since Table 2 reveals that this information was available for only 2830 PLHIV.

Reply: Thanks, corrected to Other/Unknown

Table 2: The authors divide the patients into two age groups at diagnosis (≥50 and <50). What was the rationale behind this? Usually, more age groups are analyzed. The problem is that among the four groups analyzed according to the place of birth, in three groups the IQR is below 50, which makes for even less sense of analyzing the data in this way.

Reply: We found interactions with the age dichotomy in the context of TDRM. Moreover, the effect size when these two groups were compared was larger compared to continuous age.

Table 2: The sum of PLHIV in the row “Sex, N=2819 (99%)” is not correct, it should be 631, 900, 1205, and 83. Proportions in the brackets should be corrected, accordingly.

Reply: Thanks, corrected to 631 (22.4), 900 (31.9), 1205 (42.7), 83 (2.9).

Table 2: The sum of “Transmission groups” is not 2796 but 2792, please correct.

Reply: Thanks, corrected to 2792

Table 2: In the row “A6” in the last column please correct the values, it should be “5(8.8)” and not “8.8(5)”.

Reply: Thanks, corrected to 5 (8.8).

Line 170-172: The authors state: “Interestingly, the overall proportion of any TDRM and especially of NNRTI TDRM was higher in Israeli-born individuals (8.6%) compared to those born elsewhere (3.3-7.2%, Table 2).” This statement is not correct, since overall proportion of TDRM was the highest in WCEU/NA (16.4%).

Reply: Thanks, we want to emphasize the proportion of NNRTI TDRM (the overall number of individuals from Western/Central Europe/North America is very small, 9 only, making it difficult to really conclude. We have omitted the statement of overall proportion and corrected as follows:

Interestingly, the overall proportion of NNRTI TDRM was higher in Israeli-born individuals (8.6%) compared to those born elsewhere (3.3-7.2%, Table 2).

Line 179-182: The authors state: “Two NNRTI mutations were highly prevalent: K103N/S (in 4.9%, 94/1905) and the E138AGKQ (in 5.7%, 109/1905). Both K103N/S and E138AGKQ (48 %, 22/46 and 54%, 25/46, respectively) were highly abundant in sub-subtype A1 sequences (p<0.001).” Please refer to mutations separately, since K103N/S is not a single mutation, it is two mutations, K103N and K103S. The same is true for E138AGKQ, these are four different mutations with four different mutation prevalences and also the impact of these mutations on the resistance to drugs can be different.

Table 3 and Supplementary Table S4: Please list all the detected mutations and not only per site (for example K103N and K103S separately, and not K103NS). Please correct this also in the text. The authors chose to analyze if the mutations are significantly related to the subtypes and this might have biased their results. Some mutations are known to be more frequent in some subtypes.

Reply: Thanks, Table 3 and Supplementary Table S4 were updated with separate analysis of the most frequent mutations -K103N and K103S and E138 (A,G,K,Q).Also we corrected that in the text:

"Three NNRTI mutations were highly prevalent: K103N (in 4.5%, 86/1905), E138A (in 3.7%, 70/1905) and E138Q (in 1.4%, 26/1905). Both K103N and E138Q (35%, 16/46 and 48%, 22/46, respectively) were highly abundant in sub-subtype A1 sequences (p<0.001)."

Line 197-201: The authors state: “The highest probability of having TDRM (>0.5) was observed for Israeli - born individuals below the age of 50 (OR: 2.07, 1.50-2.86 of 95% CI, p<0.001) carrying HIV-1 A1 sub-subtype (OR: 13.43, 6.57-27.52 of 95% CI, p<0.001 for male and OR: 13.84, 2.98-64.19 of 95% CI, p=0.001 for female).” This is an interesting finding, did you also check if there is any correlation with transmission risk, specifically MSM, and CD4 cell count (low vs. high; – can be somewhat an indicator of recent vs. long-term infection)?

Reply: Thank you. We have checked correlation to CD4, however, due to the rather low number of cases with available CD4 counts no correlation could be found

Discussion:

Line 232: The authors state: “Overall, more than (65%) of newly identified PLHIV were immigrants,” Please correct to “more than half (65%)”.

Reply: corrected

Line 292: The authors mention PrEP. It would be interesting to know how is the situation regarding PrEP in Israel? Can it be prescribed, is it reimbursed?

Reply: PreP has only been officially prescribed in Israel since 2017. It is not reimbursed. Therefore, we could not discuss it any further

Line 306-307: Please explain why only 52.3% had a resistance test performed, since you state that in Israel resistance testing is recommended at baseline.

Reply: This is one of the unexplained findings of this paper. We do not really understand why only half of the new cases have resistance data. Currently we are going over records of these patients in the various clinics in Israel trying to assess the impact of resistance testing at baseline (yes/no) on pts outcome. This will be analyzed and hopefully described in a new manuscript.

Line 314-316: The authors state: “In summary, the epidemiology HIV-1 in Israel is changing, shifting to individuals born in EEU/CA infected through heterosexual contacts with a reduction in the proportion of newly identified patients from SSA.” This is not correct, since the majority of individuals in this study were born in Israel and not EEU/CA. This change to EEU/CA is however seen in HIV-positive immigrants and not in the overall HIV-positive population.

Reply: Thank you for your correct comment. We have changed this sentence as follows:

" In summary, the epidemiology HIV-1 positive immigrants is changing, shifting to individuals born in EEU/CA infected through heterosexual contacts with a reduction in the proportion of newly identified patients from SSA. 

Figures:

Figures S1-3: The authors list the mutations on some of the clusters. It is however not shown, how prevalent are these mutations in the clusters and how are the detected mutations observed elsewhere, outside the highlighted clusters. This information is also lacking in the caption of these figures. What mutations were inspected? Do the authors limit themselves only on TDRM? Inclusion of more data on this topic would be interesting to see, to better observe the spread of certain TDRM in the population. The authors could consider adding an additional table to highlight the largest transmission clusters and the relevant mutation observed there, together with the demographic data. In addition, seroconversion data is presented in some of the clusters in the figures. There is no mentioning of this trait in the text. If the authors wish to include this, it should be mentioned in the Method section and in the caption of the figures. In this regard, if the data of seroconversion is included, it could have been also analyzed to see if there is any correlation between seroconversion and the presence of TDRM, etc.

Reply: Thanks, the additional table was added in Supplementary materials (Supplementary Table S6) to highlight the largest transmission clusters and the relevant DRM observed there, together with the demographic data. We also decided not to include the seroconversion data in this manuscript and analyze this theme in depth in a separate study.

Reviewer 2 Report

Manuscript Title: Epidemiology and transmitted HIV-1 drug resistance among treatment-naïve individuals in Israel, 2010-2018

Target Journal: Viruses

Manuscript ID: viruses-1535594

Manuscript Type: Article

Comments:

The authors have characterized the demographic and clinical data of people diagnosed with HIV-1 in Israel in 2010-2018 and described the epidemiology change of HIV-1 in Israel. However, the manuscript is confused and obscure. Publication is not recommended based on current form.

  1. Delete the “Background”, “Methods”, “Results” and “Conclusions” in the Abstract and reorganize the sentences.
  2. Large amounts of abbreviates would lead to confusion. Standardize the abbreviations and add a list of abbreviations.
  3. Move Supplementary Figures 1-3 to the main text.
  4. Standardize the format of the references.

Author Response

Dear reveiwer

Please find below our reply to your comments

We have addressed all issues and corrected the text accrodingly

Thank you for your contribution.

Orna

Rev 2: Comments:

The authors have characterized the demographic and clinical data of people diagnosed with HIV-1 in Israel in 2010-2018 and described the epidemiology change of HIV-1 in Israel. However, the manuscript is confused and obscure. Publication is not recommended based on current form.

  1. Delete the “Background”, “Methods”, “Results” and “Conclusions” in the Abstract and reorganize the sentences.

Reply: removed and reorganized

  1. Large amounts of abbreviates would lead to confusion. Standardize the abbreviations and add a list of abbreviations.

Reply: List of abbreviations has been included

  1. Move Supplementary Figures 1-3 to the main text.

Reply: All 3 figures moved to main text

  1. Standardize the format of the references.

Reply: The references are all in Vancouver style

Round 2

Reviewer 2 Report

Recommend to publish.